# Extracellular Vesicles in Atrial Fibrillation—State of the Art

**DOI:** 10.3390/ijms23147591

**Published:** 2022-07-08

**Authors:** Grzegorz Procyk, Dominik Bilicki, Paweł Balsam, Piotr Lodziński, Marcin Grabowski, Aleksandra Gąsecka

**Affiliations:** 11st Chair and Department of Cardiology, Medical University of Warsaw, Banacha 1A, 02-097 Warsaw, Poland; pawel.balsam@wum.edu.pl (P.B.); piotr.lodzinski@wum.edu.pl (P.L.); marcin.grabowski@wum.edu.pl (M.G.); gaseckaa@gmail.com (A.G.); 2Faculty of Medicine, Medical University of Warsaw, Banacha 1A, 02-097 Warsaw, Poland; bilicki.dominik@wp.pl

**Keywords:** atrial fibrillation, extracellular vesicles, microRNA, ablation

## Abstract

Extracellular vesicles are particles released from cells and delimited by a lipid bilayer. They have been widely studied, including extensive investigation in cardiovascular diseases. Many scientists have explored their role in atrial fibrillation. Patients suffering from atrial fibrillation have been evidenced to present altered levels of these particles as well as changed amounts of their contents such as micro-ribonucleic acids (miRs). Although many observations have been made so far, a large randomized clinical trial is needed to assess the previous findings. This review aims to thoroughly summarize current research regarding extracellular vesicles in atrial fibrillation.

## 1. Introduction

Atrial fibrillation (AF) is defined by the European Society of Cardiology (ESC) as a supraventricular tachyarrhythmia with uncoordinated atrial electrical activation and consequently ineffective atrial contraction [1]. Not only does it worsen the quality of the affected patients’ life [2] but also increases morbidity and mortality due to, e.g., increased risk of stroke [3]. It is currently the most common sustained cardiac arrhythmia in adults all over the world with an estimated prevalence between 2% and 4% [4].

Several risk factors for AF progression have been identified, such as age, heart failure, hypertension, chronic kidney disease, chronic pulmonary diseases, diabetes mellitus, previous stroke, and left atrial size [5]. However, it has not been explicitly established if biomarkers exhibit any added predictive value in terms of AF progression; moreover, in “2020 ESC Guidelines for the diagnosis and management of atrial fibrillation”, the role of biomarkers in AF management has been pointed out as having gaps in evidence [1]. Thus, research investigating the facility of biomarkers in AF patients is warranted.

According to the International Society for Extracellular Vesicles (ISEV), extracellular vesicles (EVs) are defined as particles naturally released from cells and delimited by a lipid bilayer; moreover, they cannot replicate, as they do not contain a functional nucleus [6]. Historically, several terms describing different types of EVs have been used, such as exosomes, ectosomes, or microparticles [7]. Nevertheless, the current scientific consensus advises against using these terms unless markers of subcellular origin can be established [6]. EVs play a crucial role in cell-to-cell communication [8]. Therefore, their importance in different diseases has been thoroughly studied by many research groups [9]. EVs have also been extensively investigated in various cardiovascular diseases, in terms of their involvement in pathophysiology or being potential biomarkers, as their concentrations can be determined with the use of the accessible techniques, e.g., flow cytometry (Figure 1) [10,11,12].

Because EVs have been suggested to play a potential role in AF [13,14,15,16,17], the purpose of this review is to summarize the current data on EVs’ diagnostic and prognostic utility in AF patients.

## 2. Extracellular Vesicles in Patients with Atrial Fibrillation

We analyzed the available literature for original scientific papers concerning EVs in AF. After a meticulous analysis of the pertinent studies, we divided them into the following groups: preclinical studies, EVs in AF patients treated with anticoagulants and other drugs, EVs containing nucleic acids in AF patients, EVs in AF patients undergoing ablation or other invasive procedures, and other research studies concerning EVs in AF patients.

### 2.1. Preclinical Studies

It was proven that cardiomyocytes, treated with the EVs derived from myofibroblasts, showed the downregulation of L-type calcium channel Cav1.2. This downregulation is considered to be a characteristic feature of the ionic remodeling associated with AF [18]. Additionally, EVs derived from the angiotensin II-treated human cardiac myocytes encouraged macrophages to conducted M1 polarization and, consequently, proinflammatory state; moreover, it was evidenced that Plasmacytoma Variant Translocation 1 (PVT1) contained in EVs promoted extracellular matrix remodeling in atrial fibroblasts [19].

Another research group established that beagles undergoing rapid atrial pacing for 7 days showed a rise in the atrial and plasma EVs release; moreover, it was effectively hampered by GW4869, a commonly used agent inhibiting EVs generation. It was also suggested that miR-21-5p could play a role, as it was upregulated in both atrial and plasma EVs [20]. EVs originating from mouse adipose-tissue-derived mesenchymal stem cells and transfected with the X-inactive specific transcript (XIST) caused a decrease in inflammation and myocardial pyroptosis [21]. In a recent study, EVs from bone marrow mesenchymal stem cells were transduced with the nuclear factor-erythroid 2-related factor 2 (Nrf2). These EVs injected into rats with AF not only shortened AF duration and reduced cardiomyocyte apoptosis but also minimized atrial fibrosis [22]. It was also proven that EVs derived from bone marrow mesenchymal stem cells overexpressing miR-148a reduced cardiomyocyte apoptosis by inhibiting SPARC-associated modular calcium-binding protein 2 (SMOC2) [23]. The preclinical studies discussed in this subsection are graphically summarized in Figure 2.

### 2.2. EVs in Atrial Fibrillation Patients Treated with Anticoagulants and Other Drugs

Chirinos et al. investigated the association between digoxin use in patients suffering from non-valvular AF and the level of EVs originating from endothelium and platelets. They showed that patients taking digoxin exhibited increased levels of endothelial EVs [24]. Lau et al. studied the population of AF patients treated with warfarin. It was shown that endothelial and platelet EVs collectively negatively correlated with the estimated glomerular filtration rate (eGFR), making them potential nephrotoxicity biomarkers. However, platelet-derived EVs alone did not present any correlation with the eGFR [25]. Lenart-Migdalska et al. studied the impact of dabigatran intake on platelet and endothelial EV concentration in non-valvular AF patients. Platelet-derived EVs were increased in patients treated with dabigatran; moreover, dabigatran concentration correlated negatively with the concentration change among these EVs. Endothelial-derived EVs did not exhibit such relationships [26]. The same research group evidenced that non-valvular AF patients treated with rivaroxaban presented increased levels of both platelet- and endothelial-derived EVs after drug administration [27]. Moreover, Weiss et al. demonstrated that non-valvular AF patients treated with rivaroxaban had a significantly altered proteomic profile of EVs compared with those treated with warfarin; proinflammatory proteins and complement factors were decreased, whereas negative regulators of inflammatory pathways were elevated in patients treated with rivaroxaban [28]. Complementary to the aforementioned studies, Duarte et al. showed that AF patients treated with rivaroxaban or warfarin presented increased levels of platelet-derived EVs compared with the control subjects with no AF. However, no differences in endothelial-derived EVs were noted [29]. Further improvement in this field may lead to the treatment individualization with EVs being potential predictors of treatment response in AF patients. All studies discussed in this subsection with additional data are summarized in Table 1.

### 2.3. EVs Containing Nucleic Acids in Atrial Fibrillation Patients

Wang et al. investigated EVs containing micro-ribonucleic acids (miRs). They proved that patients suffering from non-valvular AF presented increased levels of miR-483-5p and decreased levels of miR-142-5p, miR-223-3p, and miR-223-5p compared with controls in sinus rhythm. Moreover, miR-483-5p, miR-142-5p, and miR-223-3p were shown to be related to AF by univariate logistic analysis, whereas multivariate logistic analyses proved miR-483-5p to be independently correlated with AF [30]. Mun et al. researched differences in the expression of miRs in circulating EVs between subjects with supraventricular tachycardia and patients suffering from persistent AF. The study revealed that the latter group presented increased levels of miR-107, miR-320d, miR-103a-3p, miR-486-5p, and let-7b-5p [31]. Soltész et al. proved that there were no differences in the EV-contained mitochondrial deoxyribonucleic acid (mtDNA) copy numbers between AF patients and healthy controls [32]. Additionally, Wei et al. demonstrated that AF patients had increased levels of miR-92b-3p, miR-1306-5p, and miR-let-7b-3p contained in the EVs compared with control patients in sinus rhythm [33].

Liu et al. found that patients suffering from congenital heart disease (CHD) and AF presented different concentrations of miRs associated with EVs when compared with CHD patients in sinus rhythm. Quantitative analysis showed reduced levels of miR-382-3p and miR-450a-2-3p as well as increased levels of miR-3126-5p in AF patients [34]. Wang et al. showed that AF patients had higher expression of EV-contained miR-107 compared with healthy controls [35].

Interestingly, Liu et al. demonstrated that AF patients presented decreased expression levels of EV-incorporated LINC00636 (Long Intergenic Non-Protein Coding RNA) and miR-450a-2-3p compared with non-AF patients. Moreover, the expression levels of these two RNAs positively correlated with each other [36]. Importantly, Chen et al. proved that AF patients showed an increased expression of myocardial infarction associated transcript (MIAT) in serum-derived EVs when compared with healthy individuals. Interestingly, the highest MIAT expression was observed in patients with permanent AF. It was also evidenced that MIAT, abundant in those EVs, promoted atrial fibrosis and thus compounded atrial remodeling and subsequent AF [37].

Siwaponanan et al. evidenced that AF patients presented increased levels of miR-106b-3p, miR-590-5p, miR-339-3p, miR-378-3p, miR-328-3p, and miR-532-3p derived from EVs. These miRs were suggested to be possibly involved in processes such as arrhythmogenesis or structural remodeling in AF [38]. Similarly, Zhu et al. investigated EV-contained miRs expression levels in AF patients. They presented an increase in miR-124-3p, miR-378d, miR-2110, and miR-3180-3p levels as well as a decrease in miR-223-5p, miR-574-3p, miR-125a-3p, and miR-1299 levels when compared with patients in sinus rhythm [39]. Finally, Mun et al. proved that patients with persistent AF showed a significant downregulation of miR-30a-5p in small EVs [40]. Just as EVs alone, miRs and other small molecules contained in EVs can become not only diagnostic but also predictive tools. All studies discussed in this subsection with additional data are summarized in Table 2.

### 2.4. EVs in Atrial Fibrillation Patients Undergoing Ablation or Other Invasive Procedures

Herrera-Siklódy et al. studied the differences between AF patients undergoing cryoablation and radiofrequency (RF) ablation in terms of EVs as cellular damage markers. In both groups of patients, they observed an increase in platelet- and leukocyte-derived (but not endothelial-derived) EVs after the ablation procedure. However, there were no significant differences between the groups [41]. Jesel et al. compared the concentrations of EVs between the right and the left atria in AF patients undergoing ablation. They investigated procoagulant EVs derived from platelets, leukocytes, and endothelial cells. Interestingly, only endothelial-derived EVs presented atrial-specific differences, being increased in the right atrium [42]. Liles et al. measured the levels of tissue-factor extracellular vesicles (TF-EVs) in non-valvular AF patients prior to ablation surgery. They compared patients and healthy controls and divided patients based on the type of their anticoagulant therapy. It was shown that TF-EVs were increased in AF patients compared with healthy controls; however, after dividing AF patients into two groups, not-treated with anticoagulants and treated with anticoagulants, only the latter showed increased levels of TF-EVs compared with healthy controls. Consistently, this group also presented increased levels of TF-EVs compared with the first group. Interestingly, there were no differences in TF-EV levels between AF patients treated with warfarin and AF patients treated with apixaban or rivaroxaban [43]. Pourtau et al. proved the diminished tissue-factor-dependent procoagulant activity of EVs in both paroxysmal and persistent AF patients undergoing ablation compared with healthy controls; however, only the paroxysmal AF patients showed decreased fibrinolytic activity of EVs compared with controls. Moreover, 10 AF patients in sinus rhythm (for 10 days before ablation) were subjected to induced AF. After 20 minutes of acute AF, these patients presented decreased procoagulant activity of EVs with unaltered fibrinolytic activity [44].

Meng et al. demonstrated that AF patients had elevated levels of TF-EVs and EVs derived from platelets, endothelial cells, and leukocytes (but not from erythrocytes) compared with healthy controls. It was also proven that both endothelial-derived EVs (>355/μL) as well as leukocyte-derived EVs (>639/μL) were risk factors for the early recurrence of atrial fibrillation (ERAF). Additionally, the former were shown to be an independent predictor of the ERAF [45].

Mørk et al. analyzed EVs (particularly TF-EVs) in patients with or without AF undergoing cardiac surgery procedures. Total EVs, as well as TF-EVs, were shown to be increased in patients suffering from AF in all measurements, from venous blood preoperatively/intraoperatively and directly from the left atrial appendage intraoperatively [46]. Amabile et al. showed that non-valvular AF patients who underwent left atrial appendage occlusion presented increased levels of annexin V-positive, and platelet-, erythrocyte-, and leukocyte-derived EVs after the intervention. In patients undergoing coronary angiography, who served as a control group, only annexin V-positive EVs were proven to be increased after the procedure [47]. Shaihov-Teper et al. investigated the population of patients with or without AF undergoing elective heart surgery. Organ cultures, grown from epicardial fat, secreted more EVs in samples from AF patients. Moreover, EV contents, proinflammatory and profibrotic cytokines as well as profibrotic miR, were also elevated in this group [48]. Perhaps ongoing and future research will provide evidence that will lead to the EVs’ inclusion into the qualification criteria for invasive procedures such as ablation. All studies discussed in this subsection with additional data are summarized in Table 3.

### 2.5. Other Research Studies Concerning EVs in Atrial Fibrillation Patients

Choudhury et al. proved that both patients suffering from AF and patients in sinus rhythm suffering from other cardiovascular diseases had increased levels of platelet EVs compared with healthy controls in sinus rhythm. Moreover, they evidenced that neither AF type (paroxysmal or permanent) nor applied therapy (aspirin or warfarin) influenced the level of platelet EVs [49]. Ederhy et al. showed increased levels of annexin V-positive EVs in AF patients compared with control subjects with or without cardiovascular risk factors. The level of platelet- and endothelial-derived EVs showed no differences between AF patients and control subjects with cardiovascular risk factors; however, they were increased in those two groups as compared with control subjects without cardiovascular risk factors [50]. Azzam et al. proved that patients suffering from valvular AF had increased levels of platelet-derived EVs compared with age-matched healthy volunteers in sinus rhythm. Moreover, it appeared that the severity of mitral stenosis correlated with EV levels [51].

Wang et al. investigated non-valvular AF patients, both with paroxysmal AF and persistent AF. It emerged that the latter group had increased levels of EVs compared with normal controls and paroxysmal AF patients. What is more interesting is it was shown that patients with persistent AF had increased levels of EV-bound interleukin-1β and P-selectin [52]. Hayashi et al. compared the levels of P-selectin-positive EVs in platelet-rich plasma between non-valvular AF patients and control subjects; there were no differences between these groups. However, they proved that the induction of AF in paroxysmal AF patients resulted in increased levels of EVs expressing P-selectin [53]. Idriss et al. proved that both mitral valve disease (MVD) patients with AF and MVD patients in sinus rhythm had increased levels of EVs showing positive binding to anti-CD41a as compared with healthy controls; however, there were no differences between these two groups of MVD patients [54].

Wang et al. studied non-valvular AF patients with or without left atrial thrombi. Both groups showed increased phosphatidyl-serine (PS)-exposing EV levels compared with control subjects; moreover, the patients with thrombi had elevated EV levels compared with the patients without thrombi. Both groups presented an increased procoagulant activity to be effectively inhibited by the addition of lactadherin. Interestingly, the amount of EVs positively correlated with the thrombus diameter [55]. Siwaponanan et al. demonstrated that non-valvular AF patients had increased levels of total EVs and increased levels of platelet- and endothelial-derived EVs compared with healthy controls [56].

Thulin et al. investigated AF patients (some of whom were stroke cases) selected from a large randomized clinical trial and a cohort of the randomly selected control individuals of age 70. It was shown that AF patients presented increased levels of PS-positive EVs and EVs derived from platelets, leukocytes, and erythrocytes (but not from endothelial cells) compared with controls. Moreover, there were no differences in the EV levels among the AF patients when comparing the stroke cases to the others [57]. Wang et al. classified non-valvular AF patients as “low to moderate risk” or “high risk” of stroke using the CHADS2 score. The latter group was evidenced to have increased levels of annexin V-positive and platelet-derived EVs as compared with the lower-risk group. Moreover, it was shown that EVs derived from AF patients bound to platelet receptor CD36 and activated platelets [58].

Voukalis et al. evidenced that AF patients presented increased levels of apoptotic EVs (annexin V-positive) when compared with ischemic disease patients in sinus rhythm; however, both groups had similar levels of platelet-derived EVs [59]. Ni et al. conducted a proteomic analysis of serum EVs comparing paroxysmal AF patients with healthy subjects. These two groups were proven to have different expression levels of many proteins mainly involved in anticoagulation, complement system, and protein folding, as indicated by the bioinformatic analysis [60]. Zietzer et al. proved that patients suffering from AF had higher levels of large EVs derived from platelets in the left atrial appendage than patients with no AF. Moreover, patients with permanent AF presented higher levels of these EVs when compared with non-permanent AF patients [61]. All studies discussed in this subsection with additional data are summarized in Table 4.

## 3. Conclusions and Future Perspectives

Multiple studies concerning EVs in patients with AF have been conducted so far. EV plasma concentration changes of different origins have been found in patients suffering from AF; moreover, several correlations between EV concentrations and the type of AF or the type of anticoagulant treatment have been identified. Not only EV concentration *per se* have been found to be affected but also their cargos, including miRs and long non-coding RNAs. However, most studies have enrolled a rather small sample of patients; therefore, large randomized clinical trials evaluating the most important findings and possibly assessing other undiscovered features are needed. Although the detailed role of EVs in AF remains not fully discovered, it is surely crucial; thus, subsequent research in this field is highly required.

Further improvement of our knowledge about EVs in various diseases, particularly in AF, would undoubtedly contribute to the deeper understanding of the molecular basis and thus to better diagnostic and therapeutic strategies to deal with these conditions. Establishing the cut-off values of certain EV concentrations for AF diagnosis would be of great importance. Furthermore, the discovered correlations between EV concentration and various diseases could probably lead to the development of, e.g., directed therapy. Nevertheless, future research is needed to fill the gaps in the body of evidence indicated by the ESC concerning the role of biomarkers in AF management.

## Figures and Tables

**Figure 1 ijms-23-07591-f001:**
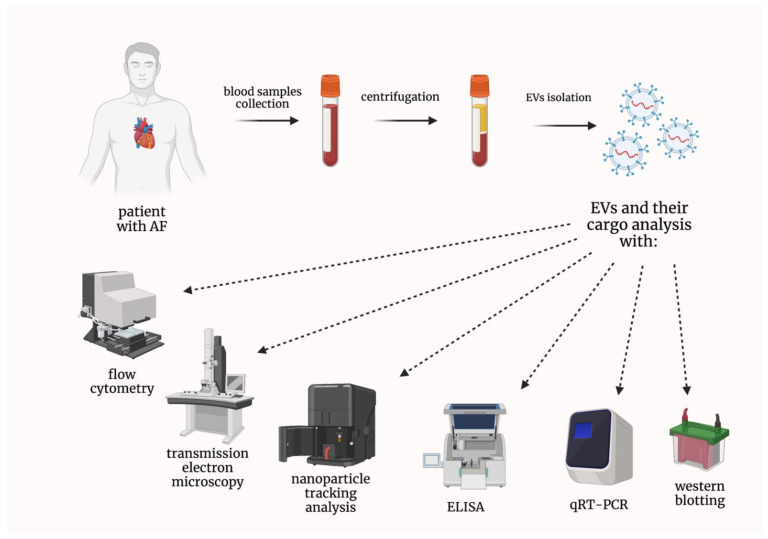
Techniques used in the analysis of both extracellular vesicles and their cargo. AF—atrial fibrillation; ELISA—enzyme-linked immunosorbent assay; EVs—extracellular vesicles; qRT-PCR—quantitative real-time reverse transcription polymerase chain reaction.

**Figure 2 ijms-23-07591-f002:**
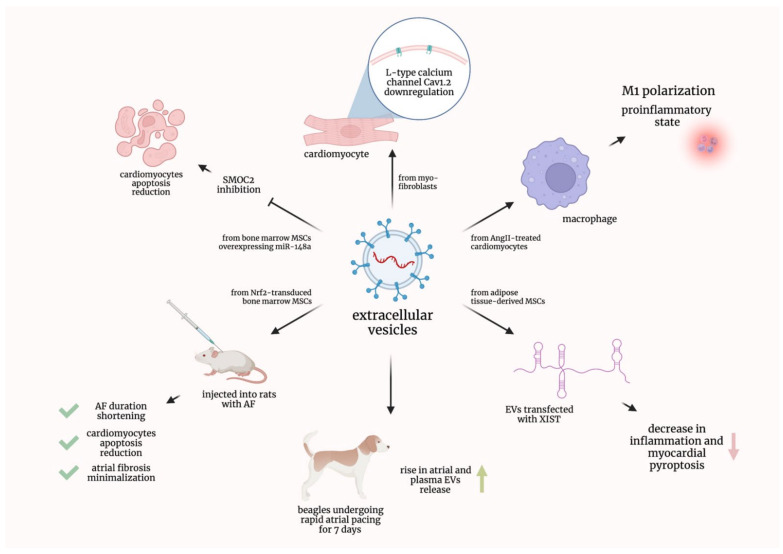
Graphical summarization of the preclinical studies concerning extracellular vesicles in atrial fibrillation. AF—atrial fibrillation; AngII—Angiotensin II; EVs—extracellular vesicles; miR—micro-ribonucleic acid; MSCs—mesenchymal stem cells; Nrf2—nuclear factor-erythroid 2-related factor 2; SMOC2—SPARC-associated modular calcium-binding protein 2; XIST—X-inactive specific transcript.

**Table 1 ijms-23-07591-t001:** The summary of recent studies regarding extracellular vesicles in atrial fibrillation patients treated with anticoagulants and other drugs.

Ref.	Population	EVs’ Origin	Comparison	Outcomes	Method
[24]	30 NV AF pts	platelets endothelial cells	pts taking digoxin vs. pts not taking digoxin	platelet EVs not significantly altered ↑ endothelial EVs in AF pts taking digoxin	EVs measured in PPP with fluorescent mAbs
[25]	160 NV AF pts treated with warfarin	platelets mixed from endothelial cells and platelets (EPEVs)	correlations with different variables	EPEVs correlated negatively with eGFR no correlations btw platelet-derived EVs and eGFR	EVs measured in PFP by FC
[26]	39 NV AF pts treated with dabigatran	platelets endothelial cells	correlations with dabigatran intake	↑ platelet EVs after taking dabigatran change in platelet EVs correlated negatively with dabigatran concentration endothelial EVs unaltered by dabigatran	EVs measured by FC
[27]	34 NV AF pts treated with rivaroxaban	platelets endothelial cells	correlations with rivaroxaban intake	↑ platelet and endothelial EVs after rivaroxaban administration	EVs measured in PFP by FC
[28]	8 NV AF pts treated with rivaroxaban	circulating EVs	15 NV AF control pts treated with warfarin	EVs of pts treated with rivaroxaban: ↓ proinflammatory proteins and complement factors ↑ negative regulators of inflammatory pathways	EVs examined by NTA, FC, and proteomics
[29]	78 NV AF pts (39 pts on rivaroxaban + 39 pts on warfarin)	platelets endothelial cells	42 control subjects in SR	↑ platelet-derived EVs in AF pts no differences in endothelial-derived EVs	EVs measured in PPP by FC

↑—increased; ↓—decreased; AF—atrial fibrillation; btw—between; eGFR—estimated glomerular filtration rate; EV—extracellular vesicle; FC—flow cytometry; mAbs—monoclonal antibodies; NTA—nanoparticle tracking analysis; NV—non-valvular; PFP—platelet-free plasma; PPP—platelet-poor plasma; pts—patients; ref.—reference; SR—sinus rhythm; vs.—versus.

**Table 2 ijms-23-07591-t002:** Summary of recent studies concerning extracellular vesicles containing nucleic acids in atrial fibrillation patients.

Ref.	Population	EV Content	Comparison	Outcomes	Method
[30]	40 NV persistent AF pts	miRs	20 controls in SR	↑miR-483-5p in AF pts ↓ miR-142-5p, miR-223-3p, miR-223-5p in AF pts miR-483-5p, miR-142-5p, miR-223-3p related with AF (univariate analysis) miR-483-5p independently correlated with AF (multivariate analysis)	EVs examined by TEM and NTA miRs levels by qRT-PCR
[31]	40 persistent AF pts	miRs	20 controls with SVT	↑ miR-107, miR-320d, miR-103a-3p, miR-486-5p, and let-7b-5p in AF pts	EVs examined by TEM and NTA miRs levels by qRT-PCR
[32]	60 AF pts	mtDNA	72 healthy controls	no differences in EV-encapsulated mtDNA copy numbers between AF pts and controls	EVs from cell-free plasma mtDNA copy number by qRT-PCR
[33]	20 AF pts	miRs	20 pts in SR	↑ miR-92b-3p, miR-1306-5p, and miR-let-7b-3p in AF pts	plasma-derived EVs miRs levels by qRT-PCR
[34]	60 adult pts with CHD and persistent AF	miRs	60 adult pts with CHD in SR	↑ miR-3126-5p in AF pts ↓ miR-382-3p and miR-450a-2-3p in AF pts	EVs from pericardial fluid miRs levels by qRT-PCR
[35]	AF pts	miRs	healthy controls	↑ miR-107 in AF pts	miRs levels by qRT-PCR
[36]	12 AF pts	RNAs	12 non-AF pts	↓ LINC00636 and miR-450a-2-3p in AF pts	EVs from pericardial fluid RNAs levels by qRT-PCR
[37]	20 AF pts	DNA	20 healthy individuals	↑ MIAT expression in EVs in AF pts	EVs examined by TEM, NTA, and Western blots qRT-PCR
[38]	30 AF pts	miRs	30 control subjects in SR	↑ miR-106b-3p, miR-590-5p, miR-339-3p, miR-378-3p, miR-328-3p, and miR-532-3p in AF pts	EVs examined by FC, TEM, NTA, and Western blots miRs levels by qRT-PCR
[39]	40 AF pts	miRs	40 pts in SR	↑ miR-124-3p, miR-378d, miR-2110, and miR-3180-3p in AF pts ↓ miR-223-5p, miR-574-3p, miR-125a-3p, and miR-1299 in AF pts	EVs examined by TEM, NTA, and Western blots miRs levels by qRT-PCR

↑—increased; ↓—decreased; AF—atrial fibrillation; CHD—congenital heart disease; EV—extracellular vesicle; FC—flow cytometry; LINC—Long Intergenic Non-Protein Coding RNA; MIAT—myocardial infarction associated transcript; miR—micro-ribonucleic acid; mtDNA—mitochondrial deoxyribonucleic acid; NTA—nanoparticle tracking analysis; NV—non-valvular; pts—patients; qRT-PCR—quantitative real-time reverse transcription polymerase chain reaction; ref.—reference; RNA—ribonucleic acid; SR—sinus rhythm; SVT—supraventricular tachycardia; TEM—transmission electron microscopy.

**Table 3 ijms-23-07591-t003:** Summary of recent studies regarding extracellular vesicles in atrial fibrillation patients undergoing ablation and other procedures.

Ref.	Population	EVs Origin	Comparison	Outcomes	Methods
[41]	60 AF pts undergoing ablation	procoagulant EVs from leukocytes, endothelial cells, platelets	cryoablation: 30 AF pts RF ablation: 30 AF pts	both groups: release of procoagulant EVs (platelet and leukocyte, but not endothelial) no significant differences btw groups	EVs measured in PPP PS content measured by FPA
[42]	22 AF pts undergoing ablation	procoagulant EVs from leukocytes, endothelial cells, platelets	comparison of EV levels btw right and left atria	↑ endothelial EVs in the right atrium no differences in total, platelet, nor leukocyte EVs	EVs measured in PPP collected in right and left atria PS content measured by FPA
[43]	30 NV AF pts prior to ablation	TF-EVs	Healthy controls NV AF pts divided into groups according to ACT	↑ TF-EVs in AF pts compared with healthy controls ↑ TF-EVs in AF pts on ACT compared with AF pts not on ACT no differences btw AF pts on warfarin and AF pts on apixaban/rivaroxaban	TF-EVs measured in plasma by ELISA
[44]	37 AF pts undergoing ablation (21 paroxysmal + 16 persistent)	fibrinolytic and TF-dependent PCA of EVs	11 healthy controls 10 AF pts in SR subjected to induced AF	↓ fibrinolytic EVs in paroxysmal AF pts compared with controls ↓ procoagulant EVs in both paroxysmal and persistent AF pts compared with controls ↓ PCA after 20 min of induced AF (unaltered fibrinolytic activity)	measurements in PFP fluorogenic assay of factor Xa chromogenic test of plasmin
[45]	56 AF pts undergoing PVI ablation	platelet, leukocyte, endothelial-cell, erythrocyte, TF-EVs	40 healthy controls	endothelial EVs > 355/μL as independent predictor of ERAF ↑ EVs of all types except erythrocyte EVs in AF pts compared with controls	EVs measured in PFP by FC
[46]	13 NV AF pts undergoing elective AVS or CABG	AnnV-positive EVs TF-EVs	12 patients in SR undergoing elective AVS or CABG	↑ total EVs in AF patients ↑ TF-EVs in AF patients	plasma EVs measured using protein microarray platform
[47]	25 NV AF pts treated with LAA occlusion	platelet, endothelial-cell, leukocyte, erythrocyte, AnnV-positive EVs	25 control patients who underwent coronary angiography	↑ AnnV-positive, platelet, erythrocyte, and leukocyte EVs after LAA occlusion ↑ AnnV-positive in controls after angiography	EVs measured in PFP by FC
[48]	30 AF pts undergoing elective heart surgery	epicardial-fat-derived EVs	32 pts in SR undergoing elective heart surgery	↑ EVs (with proinflammatory and profibrotic cytokines and profibrotic miR) in cultured explants from AF pts	EVs analyzed by FC, NTA, ELISA of organ cultures

↑—increased; ↓—decreased; ACT—anticoagulant therapy; AF—atrial fibrillation; AnnV—Annexin V; AVS—aortic valve surgery; btw—between; CABG—coronary artery bypass grafting; ELISA—enzyme-linked immunosorbent assay; ERAF—early recurrence of atrial fibrillation; EV—extracellular vesicle; FC—flow cytometry; FPA—functional prothrombinase assay; LAA—left atrial appendage; min—minutes; miR—microribonucleic acid; NTA—nanoparticle tracking analysis; NV—non-valvular; PCA—procoagulant activity; PFP—platelet-free plasma; PPP—platelet-poor plasma; PS—phosphatidylserine; pts—patients; PVI—pulmonary vein isolation; ref.—reference; RF—radiofrequency; SR—sinus rhythm; TF—tissue factor.

**Table 4 ijms-23-07591-t004:** Summary of other research studies concerning extracellular vesicles in atrial fibrillation patients.

Ref.	Population	EVs Origin	Comparison	Outcomes	Methods
[49]	70 NV AF pts (35 paroxysmal + 35 permanent)	platelets	46 pts in SR + 33 healthy people in SR	↑ EVs in AF pts and SR pts compared with healthy SR	EVs measured in PPP by FC
[50]	45 NV AF pts (permanent or persistent)	platelet, endothelial-cell, AnnV-positive EVs	45 control subjects with CV RFs + 45 control subjects without CV RFs	↑ AnnV-positive EVs in AF pts ↑ platelet and endothelial EVs in both AF pts and control subjects with CV RFs compared with control subjects without CV RFs	EVs measured in PPP wells coated with antibodies
[51]	20 valvular AF pts with chronic rheumatic MS	platelets	10 healthy volunteers in SR	↑ platelet EVs in AF pts correlation btw the severity of MS and EV levels	EVs measured by FC
[52]	53 NV AF pts (37 persistent + 16 paroxysmal)	EVs captured with mAbs	37 normal controls	↑ EVs in persistent AF pts compared with normal controls and paroxysmal AF pts	EVs measured in serum amount of EVs by LAP activity
[53]	28 NV AF pts	EVs expressing P-selectin	13 control subjects	↑ EVs in paroxysmal AF pts after induction of AF no differences in EVs level btw chronic AF pts and control subjects	EVs measured in PRP by FC
[54]	21 AF MVD pts	EVs with positive binding to anti CD41-a	20 healthy controls 24 MVD pts in SR	↑ EVs in both MVD pts groups compared with healthy controls (no differences btw MVD pts groups)	EVs measured in PPP by FC as percentage of total platelet count
[55]	78 NV AF pts (18 with left atrial thrombi and 60 without)	PS-positive EVs from platelets, leukocytes, erythrocytes, endothelial cells	36 matched controls	↑ EVs in both groups of NV AF pts compared with controls ↑ EVs in pts with left atrial thrombi compared with pts without thrombi PCA inhibition by blockade of exposed PS on EVs with lactadherin positive correlation btw thrombus diameter and amount of EVs	EVs measured by FC PCA evaluated with clotting time, factor Xa, thrombin, and fibrin formation
[56]	66 NV AF pts (31 permanent + 28 paroxysmal + 7 persistent)	platelet, leukocyte, endothelial-cell, erythrocyte, and PS-positive EVs, TF-EVs	33 healthy individuals	↑ total EVs and EVs from platelets and endothelial cells in AF pts compared with controls	EVs measured in centrifuged serum by FC
[57]	836 AF pts (of which 280 pts with ischemic stroke or systemic embolism)	platelet, leukocyte, erythrocyte, endothelial-cell, PS-positive EVs	1007 randomly selected 70 yo control individuals	↑ EVs (of all origins except endothelial derived) in AF pts compared with control individuals similar EV levels among all AF pts (stroke cases vs. others)	EVs measured by FC and SP-PLA
[58]	210 NV AF pts (paroxysmal or persistent)	platelets AnnV-positive	35 healthy controls pts divided in “low to moderate risk” or “high risk” of stroke	↑ AnnV-positive and platelet EVs in pts at “high risk” EVs from NV AF bound to platelet CD36 and activated platelets	EVs measured in PFP by ELISA
[59]	47 AF pts	platelets AnnV-positive	39 IHD pts in SR	↑ AnnV-positive EVs Similar levels of platelet-derived EVs	EVs measured in PPP by FC
[60]	15 paroxysmal AF pts	CD9- and CD63-positive EVs	15 healthy donors	altered expression of proteins involved in anticoagulation, complement system, and protein folding	EVs examined with TEM and Western blots quantitative proteomic analysis
[61]	48 AF pts (27 permanent AF + 21 non-permanent AF)	large EVs from platelets	10 pts with no AF	↑ platelet-derived EVs in AF pts ↑ platelet-derived EVs in permanent AF compared with non-permanent AF	PFP from LAA EVs examined by FC, TEM, NTA

↑—increased; ↓—decreased; AF—atrial fibrillation; AnnV—Annexin V; btw—between; CD—cluster of differentiation; CV—cardiovascular; ELISA—enzyme-linked immunosorbent assay; EV—extracellular vesicle; FC—flow cytometry; IHD—ischemic heart disease; LAA—left atrial appendage; LAP—leucine aminopeptidase; mAbs—monoclonal antibodies; MS—mitral stenosis; MVD—mitral valve disease; NTA—nanoparticle tracking analysis; NV—non-valvular; PCA—procoagulant activity; PFP—platelet-free plasma; PPP—platelet-poor plasma; PRP—platelet-rich plasma; PS—phosphatidylserine; pts—patients; ref.—reference; RF—risk factor; SP-PLA—solid-phase proximity ligation assay; SR—sinus rhythm; TEM—transmission electron microscopy; TF—tissue factor; yo—years old.

## Data Availability

Not applicable.

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
