# Peer review of "Extracellular Vesicles in Atrial Fibrillation—State of the Art"

_ijms, 2022, doi:10.3390/ijms23147591_

Round 1
Reviewer 1 Report
This manuscript "Extracellular vesicles in atrial fibrillation – state of the art" by Procyk et al describes the role of extracellular vesicles in Atrial Fibrillation. In this study authors have summarized the recent study related to EV and AF. Authors have wisely described the recent preclinical studies. Furthermore authors have listed the recent studies related to AF patients treated with anticoagulants. As miRNA is one of the important constituents of EV Which plays a major role in AF pathogenesis and authors have covered that part too. At last they have concluded EV relation with the patient who underwent ablation and other procedure. This manuscript is very well written and easy to understand. Authors have nicely put together the important discoveries conducted in recent years. I believe this manuscript will definitely benefit the scientist working in this field. I just have couple of minor concern 1. first is about the conclusion section. The Conclusion section's First Paragraph should be rewritten with better clarity. There are some incomplete sentences such as line 288. 2. I think it would be very wise to include a separate section (may be before conclusion section) which talks about the future possibility to utilize EV as Therapy or Diagnosis, that would give other researchers a path to future research in AF.
Thank You
Reviewer 2 Report
This is a comprehensive well-written review on the research on EVs in atrial fibrillation.
Regarding EVs, authors are aware and conform to MISEV.
Few minor points:
- Harmonize miRNA or miR
Lines 14 ≠ 83-84 ≠ 160 ≠ 214,
- In general, et al. and line 288 per se are in italic.
I recommend to publish this paper.
Reviewer 3 Report
In the present paper entitled: “Extracellular vesicles in atrial fibrillation-state of the art” by Procyk et al. the authors try to summarize all the data on EVs diagnostic and prognostic utility in atrial fibrillation patients. To this aim they divided the available literature in five subgroups. Authors’ choice certainly facilitate paper reading. The subject of this review is really interesting and actual. Unfortunately, the authors limit themselves to making a description of the published works without drawing any relevant conclusions. Although, as indicated in the final paragraph, no definitive conclusions can be drawn due to the low number of patients recruited in the various studies, the authors should better underline EV role in AF. Moreover, at the end of each section the authors should try to correlate experimental data with either disease outcome or drug treatment.
Round 2
Reviewer 3 Report
The authors improved the quality of the paper by modifying it according to reviewer suggestions.